

# A comparison of lodgepole and spruce needle chemistry impacts on terrestrial biogeochemical processes during isolated decomposition

Laura T. Leonard[1], Kristin Mikkelson[1], Zhao Hao[2], Eoin L. Brodie[2], Kenneth H. Williams[2,3] and Jonathan O. Sharp[1,3]

[1] Colorado School of Mines, Golden, CO, USA
[2] Lawrence Berkeley National Laboratory, Berkeley, CA, USA
[3] Rocky Mountain Biological Laboratory, Crested Butte, CO, USA

## ABSTRACT

This study investigates the isolated decomposition of spruce and lodgepole conifer needles to enhance our understanding of how needle litter impacts near-surface terrestrial biogeochemical processes. Harvested needles were exported to a subalpine meadow to enable a discrete analysis of the decomposition processes over 2 years. Initial chemistry revealed the lodgepole needles to be less recalcitrant with a lower carbon to nitrogen (C:N) ratio. Total C and N fundamentally shifted within needle species over time with decreased C:N ratios for spruce and increased ratios for lodgepole. Differences in chemistry correlated with $CO_2$ production and soil microbial communities. The most pronounced trends were associated with lodgepole needles in comparison to the spruce and needle-free controls. Increased organic carbon and nitrogen concentrations associated with needle presence in soil extractions further corroborate the results with clear biogeochemical signatures in association with needle chemistry. Interestingly, no clear differentiation was observed as a function of bark beetle impacted spruce needles vs those derived from healthy spruce trees despite initial differences in needle chemistry. These results reveal that the inherent chemistry associated with tree species has a greater impact on soil biogeochemical signatures during isolated needle decomposition. By extension, biogeochemical shifts associated with bark beetle infestation are likely driven more by changes such as the cessation of rhizospheric processes than by needle litter decomposition.

## INTRODUCTION

Forest litter decomposition alters surrounding soil by influencing mechanisms that include terrestrial carbon (C) and nitrogen (N) cycling (*Šantrůčková, Krištůfková & Vaněk, 2006*; *Hicks Pries et al., 2017*). Several studies have looked further into these contributions to soil biogeochemistry; however, the observed impacts of decomposition on soil biogeochemistry differ between studies as a function of forest structure and tree

Corresponding author
Jonathan O. Sharp,
jsharp@mines.edu

mortality levels that fluctuate with time (*Negrón & Cain, 2018*; *Fraterrigo, Ream & Knoepp, 2018*). Recent bark beetle outbreaks in association with climate change have resulted in increased tree mortality within forests and pulsed litterfall that collectively result in short and long-term impacts to the forest ecosystem (*Edburg et al., 2012*; *Mikkelson et al., 2013*). A discrete analysis of litter decomposition could help us better understand its relative contribution to broader processes such as why certain forests reveal different C and N signatures from others.

A unique aspect of bark beetle tree mortality is a pulsed release of conifer needles to the forest floor that adds additional organic inputs to the litter decomposition process (*Edburg et al., 2012*). In concert, reduced canopy cover causes shifts in tree hydrologic processes such as canopy interception and evapotranspiration (*Mikkelson et al., 2013*; *Bearup et al., 2014*) further altering the decomposition process with changes to soil temperature and moisture levels. This litter decomposition is coupled to the cessation of rhizospheric processes and collectively results in tree-scale biogeochemical shifts in net soil respiration (*Brouillard et al., 2017*), soil nutrient levels (*Clow et al., 2011*; *Cigan et al., 2015*), and soil carbon concentration and character (*Brouillard et al., 2017*; *Fraterrigo, Ream & Knoepp, 2018*). However, we do not fully understand how needle decomposition vs other variables such as cessation of rhizodeposits and reduced canopy alters the magnitude, character and timing of the biogeochemical response. Understanding the contributions of increased litter inputs to soil C and N cycling is important as endemic bark beetle species are prevailing at higher magnitudes globally. Increased outbreaks in conifer forests have been documented in Europe, Canada, and the United States causing concern for forest aesthetics and economics as well as altered ecosystem and watershed function (*Natural Resources Canada, 2013*; *U.S. Forest Service, 2019*; *Hlásny et al., 2019*).

To this end, we designed a study to isolate spruce and lodgepole needle litter from the complexity of tree-scale processes. We hypothesized that the isolated decomposition of needles containing different organic carbon and nitrogen signatures would impart unique soil biogeochemical responses. Soil respiration, C, N, porewater composition, and microbial ecology were then compared as a function of tree species and bark beetle impact over 2 years of decomposition in a subalpine meadow environment (3,170 m above sea level) in Crested Butte, Colorado. Recently, mountain pine beetle (*Dendroctonus ponderosae*) outbreaks were cause for concern in Colorado with peak lodgepole pine mortality occurring in 2008 (*Colorado State Forest Service, 2018*). Spruce beetle (*Dendroctonus rufipennis*) infestation has intensified since 2011. This approach allowed us to decouple the effects of varying litter decomposition from the tree canopy and root network to better understand selective pressures exerted by needle decomposition on tree-scale biogeochemical processes and their relevance to ecosystem disruption.

## MATERIALS AND METHODS

### Needle sample collection

Needle litter samples were collected in August 2016 from separate forests containing two tree species, lodgepole pine (*Pinus contorta*) and Engelmann spruce (*Picea engelmannii*). Throughout the paper these collected needles will be referred to as "harvested needles".

The lodgepole pine needles were collected in White River National Forest near Frisco, Colorado (39.54°N, 106.15°W, elevation 3,050 m). The forest is mostly lodgepole pine and experienced mountain pine beetle infestation between 2007 and 2008 (*Brouillard et al., 2017*). The soil type of this location is a Leighcan family-Cryaquolls complex with metamorphic gneiss bedrock (*Natural Resources Conservation Service, United States Department of Agriculture, 2019*; *U.S. Geological Survey, 2005*). This site was chosen for its large area of lodgepole pine trees that have not been affected by anthropogenic activities. Litter was collected under lodgepole trees that had naturally senesced. Throughout the paper, these needles will be referred to as "lodgepole."

Spruce needles were collected from Monarch Pass (38.50°N, 106.33°W, elevation 3,440 m) located in central Colorado between Gunnison and Chaffee county where the forest is predominantly Engelmann spruce. The bedrock of this location is granitic igneous intrusive (*U.S. Geological Survey, 2005*). The soil type of this location is not available. This site was chosen for its abundance of spruce trees with active spruce beetle infestation that intensified in the area during 2015 (*Colorado State Forest Service, 2016*). Visual observations included beetle boreholes, intensive sap release from pitch tubes along with live larvae and adult beetles underneath the outer bark layers. Within months of observing beetle infestation, the impacted spruce needles turned red, further confirming spruce beetle infestation. By August 2016 the majority of needles had fallen from the impacted trees and were then collected from both impacted and naturally senesced spruce trees from this forest. The needles collected under the impacted trees are referred to as "impacted" spruce needles throughout this paper and the needles collected under non-impacted naturally senesced spruce trees will be referred to as "healthy" spruce needles.

The harvested needles were stored in a dark and dry location at room temperature to air dry until deployment. The dry needles were sieved using a Tyler Equivalent 10 mesh, 1.7 mm to homogenize the collected litter from each respective sample type and remove rocks and dirt. The sieved needles were then weighed into 250 g subsets based on the three needle sample types and aliquoted for deployment at the study site described below. All litter composition results are presented based on air-dried weight.

## Study site and needle deployment

The experimental plot used in this study is located in Crested Butte, Colorado. The plot is located within the subalpine region of the Washington Gulch drainage into the East River watershed located at 38.95°N, 107.03°W, with an elevation of 3,170 m and a northeast aspect and average slope of 9–12°. The plot is located in an open meadow to ensure isolation from surrounding tree canopies and roots. The subalpine meadow soil type is classified as a mixture of Tilton sandy loam and Cryaquolls and Histosols with sedimentary clastic Mancos shale bedrock (*Natural Resources Conservation Service, United States Department of Agriculture, 2019*; *U.S. Geological Survey, 2005*). Conifer tree stands are within approximately 300 m of the experimental plot.

Climate data was collected from the nearest snow telemetry (SNOTEL) weather station on Mt. Crested Butte (38.89°N, 106.95°W) at 3,100 m elevation (site number 380).

Mean annual precipitation over the study years of 2016–2018 averaged 65 cm, most of which falls as snow starting in September/October and continues through May. The average daily temperatures recorded at the Butte SNOTEL location were separated into the snow-free months of May–September and snow-covered months of October–April. The temperatures averaged 11.6 °C and −1.1 °C, respectively. Hourly soil temperature was measured with unshielded microclimate sensors buried on-site at 12 cm soil depth. The mean soil temperatures recorded over 2016–2018 were 12.6 °C during the months of May–September, and 1.4 °C during the months of October–April.

Needles were deployed at the experimental plot in October 2016 in 25.4 cm diameter white polyvinyl chloride (PVC) collars before winter snowfall. The collars were 18 cm in height with a beveled edge at the end. Beveled end first, the collars were driven into the ground to leave approximately 10 cm of the height above ground level. Foliage was raked and pulled out of the ground to remove all native plants and roots. A total of 16 PVC rings were inserted into the ground to hold four sample types in quadruplicate. The plot matrix was established with quadruplicate samples as follows: 4 × 250 g of healthy spruce, 4 × 250 g impacted spruce, 4 × 250 g lodgepole and 4× needle-free controls (bare soil). Randomization of harvested needle samples was achieved using R Studio to produce a 4 × 4 matrix with sample assignments. A schematic of this field deployment grid with the randomized samples can be found in Fig. S1.

Soil Moisture's 1905L06 15 cm lysimeters were installed in the middle of each ring with a bentonite seal over the surface. The needles were placed on top of the soil surface and surrounding the lysimeter. Each collar top was covered with deer netting and secured to minimize interference from native animals. Onset HOBO Data Loggers soil moisture (S-SMD-M005) and temperature (S-TMB-M006) probes were installed per manufacturer instructions at 12 cm depth within the sample grid to monitor soil moisture and temperature hourly.

While the needle deployment limited emergent plant growth, any new growth was manually pulled out of the soil with as little disturbance as possible. Frequent monitoring of the plots similarly ensured that bare soil control rings were free of emerging plant growth. In October 2017, one quadruplicate ring from each of the four sample types was sacrificed for soil extractions. This reduced the number of sample replicates to three for subsequent samplings. In addition, shaded controls were added in May of 2018 prior to any 2018 measurements with randomized placement of three PVC rings on bare soil with two layers of permeable 50% black mesh shade fabric secured over the rings to understand the impact of solar radiation (Fig. S1). Biogeochemical attributes of needle decomposition were then measured with soil respiration, porewater composition, and microbial analysis as a function of tree species and bark beetle impact over 2 years of decomposition.

## Needle composition: total carbon, nitrogen and FTIR analyses

An initial subset of each harvested needle type (healthy spruce, impacted spruce, and lodgepole) was used for compositional analysis at the start of the experiment. This same compositional analysis was conducted on a subset of needles collected from each sample

collar two years later (October of 2018). Needles were set in a dark and dry location at room temperature to air dry. Once air-dried, the needles were ground to a fine powder and submitted in triplicate for total carbon and nitrogen analysis at Colorado State University's Ecocore facility with the LECO TruSpec CN analyzer (LECO Corporation, St. Joseph, MI, USA) and Fourier Transform Infrared (FTIR) analysis at Lawrence Berkley National Laboratory. The FTIR spectrometer used was equipped with an attenuated-total-reflection accessory (Nicolet IS50, Thermo Fisher Scientific Inc., Waltham, MA, USA). The sample was pressed down uniformly on the top surface of the crystal and the reflected infrared signal from the sample was collected to a deuterated-triglycine sulfate detector. All collected absorption spectra were further preprocessed for baseline correction and peak-by-peak analysis was applied to quantify the concentration of each functional group which was linearly correlated to the absorbance following Lambert–Beer's Law.

In addition to compositional analysis, water extraction tests were conducted with a subset of the harvested needles to determine the initial extractable carbon and nitrogen concentrations. Measurements of extractable dissolved organic carbon (DOC), $UV_{254}$, and total nitrogen (TN) were conducted after equilibrating in deionized water following 1 h of shaking in 1:8 solid mass: liquid volume ratio. The extractions were filtered through 0.45 μm filters. Filtrate DOC and TN were tested using a Shimadzu TOC-550A Total Organic Carbon Analyzer after acidifying with hydrochloric acid. $UV_{254}$ was measured using a DU 800 spectrophotometer. Specific UV absorbance (SUVA) was calculated by normalizing the $UV_{254}$ values with the respective DOC concentrations in mg/L for porewater samples and mg/g for the soil and litter extractions following Environmental Protection Agency Method 415.3 and (Eq. 1) (*Potter & Wimsatt, 2009*):

$$SUVA(L/mg\text{-}m) = UVA(cm^{-1})/DOC(mg/L) \times 100 \text{ cm/m} \tag{1}$$

## Gas flux analysis

Gas flux measurements were conducted above each soil collar during the snow-free months for a total of eleven sampling events over 2 years. Data was collected over an approximate 2-h window that fell midday between 10 AM and 3 PM. The collar sampling sequence was randomized each time to account for temporal fluctuations in variables such as temperature and moisture. Gas flux measurements were conducted using a Picarro G2508 cavity ring-down spectroscopy analyzer capable of analyzing $CO_2$, $CH_4$, $NO_2$, $NH_3$ and $H_2O$. Guaranteed spec ranges of the G2508 are 0.3–200 ppm $NO_2$, 1.5–12 ppm $CH_4$, 380–5,000 ppm $CO_2$, 0–300 ppb $NH_3$, and 0–3% $H_2O$. A closed system was implemented with two lengths of norprene tubing and an airtight PVC chamber to circulate the headspace for the duration of the sample period. One length of the tubing was attached to the Picarro gas inlet and the other was attached to the outlet. Each end of the tubing was then connected to an inlet and outlet valve in the PVC chamber that was placed over the permanently deployed PVC collars. After steady state was established by linear trends of production or removal, data collection was initiated for 2 min. The Hutchinson & Mosier method provided with the Picarro computer software was used to compute a best fit for each gas trend to calculate the fluxes.

## Porewater and soil extraction analysis

Porewater was collected from within each PVC collar after snowmelt in the spring and high moisture events during early fall for a total of four collections over 2 years. Porewater was collected from the lysimeters using a Luer-lok 50 mL vacuum syringe. The samples were filtered to 0.45 µm and frozen the same day of collection for storage at −20 °C until analysis. The samples were subsequently thawed for TN, DOC, and SUVA analyses as specified in "Needle Composition: Total Carbon, Nitrogen and FTIR Analyses".

Extractions were conducted with soil underlying the needles from single quadruplicate collars sacrificed for each sample type in the Fall of 2017. The extractions were conducted according to the methods outlined in "Needle Composition: Total Carbon, Nitrogen and FTIR Analyses" except with a 2:9 solid mass: liquid weight ratio. Soil samples were collected from the upper surface horizon in contact with decomposed matter (2–3 cm) and mixed for a homogenized sample from which five subsamples were collected. Geochemical extractions were conducted with DI water following the methods of *Brouillard et al. (2017)*. Resulting filtrate at 0.45µm was analyzed for TN, DOC, and SUVA as described in "Needle Composition: Total Carbon, Nitrogen and FTIR Analyses". Ammonium concentrations were measured using the sodium salicylate method and absorbance at 650 nm with a DR 3900 Hach Spectrometer. Nitrite and nitrate were measured using an ICS-5000 ion chromatography analysis consisting of an ICS-5000 DC Conductivity Detector, an ICS-5000 DP Isocratic Pump, and an AS-DV Autosampler. All soil extraction results were normalized by air-dried weights.

## DNA extraction and preparation

Soil samples for DNA extractions were collected within each PVC collar for microbial analysis at five time points over 2 years: August 2017, October 2017, May 2018, July 2018, and October 2018 to align with the early, mid, and late-season snow-free months. Samples were collected in the upper 1–2 cm soil layer under each representative needle type with sterilized scoopulas. Homogenization was achieved by collecting a minimum of ten subsamples of soil from the first 2 cm of the upper soil surface to fill a 2 mL test tube. This process was conducted throughout the full collar surface area to completely represent the sample. Care was taken to ensure samples were randomly collected across the soil surface and not biased towards the inside or outside edges of the collar. The soil samples were then frozen for storage at −20 °C until analysis. DNA was extracted from the samples using the ZymoBIOMICS DNA Miniprep kit according to the manufacturer instructions using 0.25 g weighed out from each 2 mL sample tube.

16S and 18S gene amplification was conducted using 5 Prime Hot MasterMix and a primer set utilizing the 515-Y forward primer and 926R reverse primer (*Parada, Needham & Fuhrman, 2016*). An adapted forward primer was utilized with the addition of the M13 sequence to allow sample barcoding during PCR (*Caporaso et al., 2012*; *Stamps et al., 2016*). Resulting amplicons were purified and normalized to equimolar concentrations. The samples were then concentrated using the 30 K ultra centrifugal Amicon filters. The final concentrations were determined using the Qubit 2.0 fluorometric quantitation. The library was sent to the Duke Center for Genomic and Computational

Biology for Illumina MiSeq sequencing using V2 PE250 chemistry. Raw sequence data has been deposited in the NCBI SRA database under SRA accession number PRJNA605259.

## Sequence processing

The resulting raw reads from sequencing were joined, quality filtered, clustered and had chimeras removed using the DADA2 package (*Callahan et al., 2016*). Sequences were trimmed to excise forward and reverse primer sequences in methods utilized by *Honeyman, Day & Spear (2018)* in which the first 40 nucleotides of the forward primer sequences and the last 20 of the reverse primer were removed and then trimmed using a quality score of 2. 18S sequences were analyzed separately from 16S where the forward and reverse reads were combined by concatenating the reads directly, despite the absence of an overhanging region of 18S gene sequenced by both the forward and reverse reads. Both 18S and 16S taxonomic assignments were created using Silva v128 (*Pruesse, Peplies & Glöckner, 2012*). Sequences for each sample were filtered into either Eukaryota (18S) or Bacteria (16S). Chloroplasts and Mitochondria were removed from all 16S sequences. After processing and quality filtering for the 16S genes, 796,847 sequences were obtained across 69 samples with sequence depth ranging from 18 –19,498 sequences per sample and were rarefied to 5,638 sequences. The yield for 18S sequences with these primers was lower at 34,009 across 69 samples. Sequence depth ranged from 0 to 2,950 sequences per sample and were rarefied to 200 sequences. All subsequent analysis except differential abundance testing was based on the rarefied sequence tables.

All downstream analysis was conducted in R (v1.0.153), using the Phyloseq (v1.26.1) (*McMurdie & Holmes, 2013*), vegan (*Oksanen et al., 2019*), DESeq2 (*Love, Huber & Anders, 2014*), and ampvis2 (*Andersen et al., 2018*) packages.

## Statistical analyses

Geochemical parameters between sample types were statistically differentiated using the Friedman test followed by the post hoc Wilcoxon test. Due to low sample sizes ($n = 3$) no significant values were returned with these methods for the various geochemical tests conducted. Spearman and Pearson tests were used to assess nonlinear and linear correlations between gas flux and soil parameters. For the DNA sequence results, beta diversity was assessed using both weighted and unweighted UniFrac distance matrices and significant clustering was determined using the Adonis test. Differential abundance comparisons were conducted using DESeq2. In all statistical tests, *p*-values less than 0.05 were considered significant.

## RESULTS

### Chemical composition of harvested and decomposed needles

The initial chemical differences between needle types were determined with the harvested needles prior to deployment (Figs. 1A and 1B). Clear differences in needle chemistry were apparent. Harvested naturally senesced lodgepole needles contained the highest percentages of total C and N in contrast to both spruce needle types. The harvested impacted (beetle-killed) spruce needles were higher in total C and N than the healthy

(naturally senesced) spruce needles (Table S1). Impacted spruce exhibited a higher carbon to nitrogen (C:N) ratio in contrast to the lower ratios between the lodgepole and healthy spruce needles (Fig. 1A). After 2 years of decomposition, the differences between needle types were amplified with transitions as a function of tree species (Figs. 1A and 1C). The average total carbon content decreased for all decomposed needle types, however with high variability in the healthy spruce (39% ± 15) (Table S1). Interestingly, the average total nitrogen content increased in both impacted and healthy spruce needles while lodgepole decreased in total nitrogen, again with high variability in healthy spruce (0.9% ± 0.3) (Table S1). Overall, there was a decrease in the average C:N ratio for both impacted and healthy spruce needles and an increase in lodgepole after two years of decomposition (Fig. 1A).

FTIR analysis further confirmed compositional differences between initial harvested needle types based on several peak absorbances (Fig. 1B). Notably, the spectra for the 2016 lodgepole needles exhibited clear differences from the spruce needles in litter quality based on integrated peak areas at ether linkages (1,150 cm$^{-1}$), aromatics (1,510 cm$^{-1}$), amides (1,600 cm$^{-1}$), and carbonyls (1,720 cm$^{-1}$). Previous studies have associated lignin content with the aromatics peak and cellulose content with ether linkages (*Pandey & Pitman, 2003*; *Özgenç, Durmaz & Kuştaş, 2017*). By applying the same inferences as these past studies, differences in the aromatic and ether linkage peak areas indicate lower recalcitrance, or lignin quantity and more labile carbon such as cellulose in lodgepole needles when compared to the spruce needles (Figs. 2A and 2B).

Impacted vs healthy spruce FTIR spectra, while more similar, exhibited differences in aromatics, amides, and carbonyl peak areas. After 2 years of decomposition, integrated peak analysis revealed a shift between the 2016 harvested needles and the 2018 decomposed needles (Fig. 2). The observed differences between lodgepole and spruce needles in the four major functional groups were maintained 2 years after decomposition. Specifically, lodgepole needles maintained a more labile and less recalcitrant litter quality in contrast to both impacted and healthy spruce needles (Figs. 2A and 2B). Further, the ratio of ether linkages to aromatics (1,150:1,510 cm$^{-1}$) over 2 years of decomposition revealed a significant increase in lodgepole compared to the lesser increases in the healthy and impacted spruce needles (Fig. 2E; Table S2).

DI water extractions with the harvested needles revealed water extractable constituents differed between all needle types in which impacted spruce needles released the highest extractable carbon in contrast to lower values for lodgepole and healthy spruce needles. The total extractable nitrogen was highest in association with spruce needles in contrast to lodgepole with similar values between impacted and healthy spruce. As determined by specific UV absorbance, more aromatic organic carbon was extracted from the healthy spruce and lodgepole in contrast to the impacted spruce (Table 1).

## Gas flux from needle and control deployments

Needle presence and seasonality impacted soil gas flux over the period of the study (Figs. 3A and 3B). The presence of decomposing needles increased $CO_2$ production in contrast to the controls. Throughout all sampling events, the needle-free controls

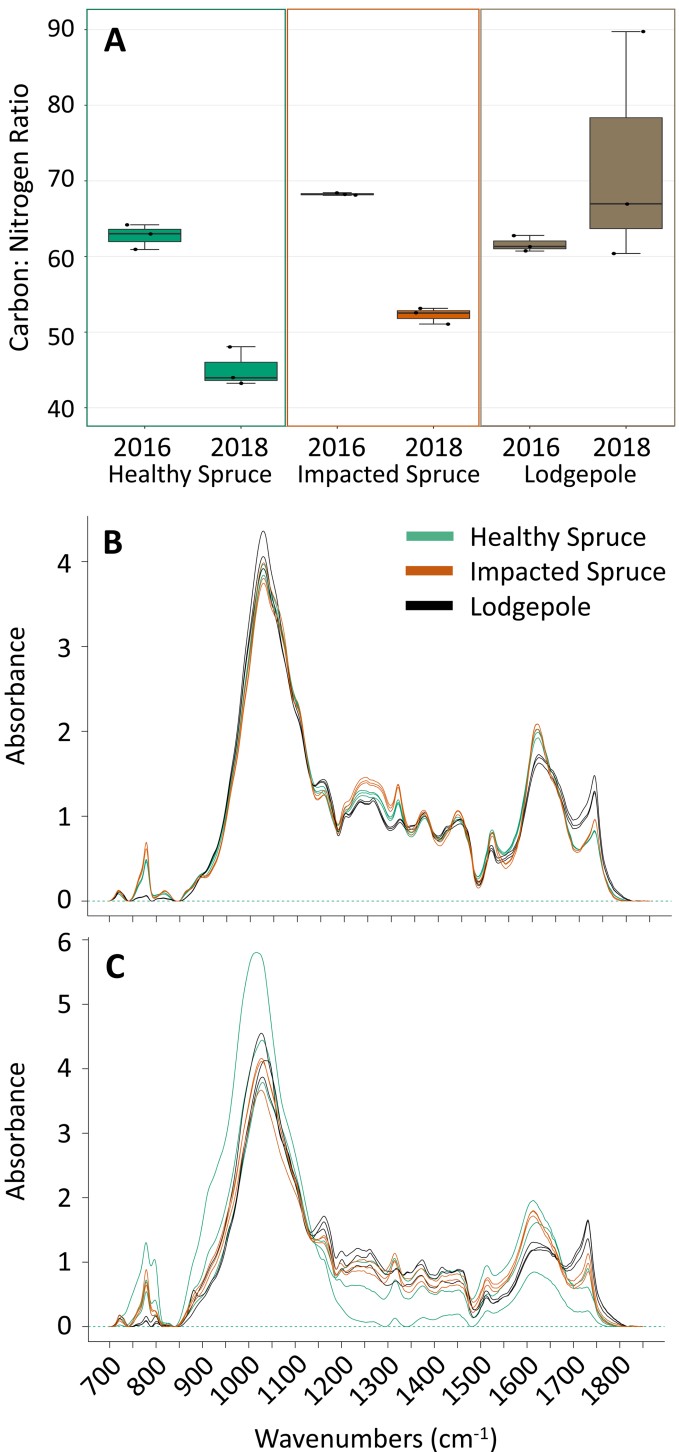

**Figure 1  Differences in needle carbon, nitrogen, and organics composition.** (A) Percentages of carbon and nitrogen needle composition represented as the carbon:nitrogen ratio ($n = 3$) for the harvested needles (2016) and after 2 years of decomposition (2018). Points on the plot represent actual values. (B) FTIR spectra for the initial needles and (C) after 2 years of decomposition. Peak heights at specific wavenumbers reveal differences in the spectra as a function of needle species. Total carbon and nitrogen percentages can be found in Table S1.                           
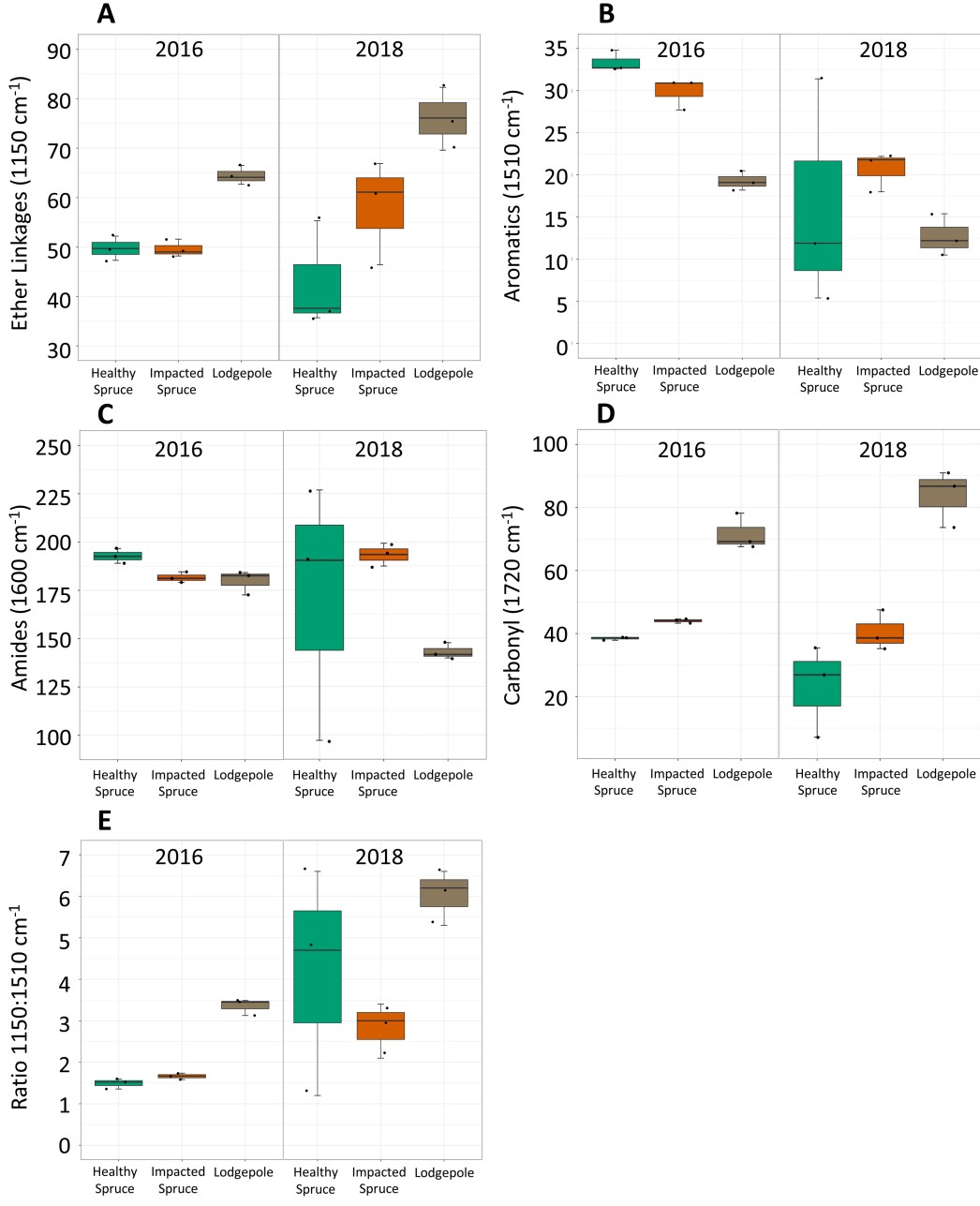

**Figure 2 Differences in integrated FTIR spectra peak areas.** Integrated areas of peak absorbance at wavenumbers with clear differences for the initial needles (2016) and after 2 years of decomposition (2018). Peak areas suggest differences in the chemical composition between needles, notably (A) ether linkages (1,150 cm$^{-1}$), (B) aromatics (1,510 cm$^{-1}$), (C) amides (1,600 cm$^{-1}$), and (D) carbonyl signatures (1,720 cm$^{-1}$). (E) The ratio of ether linkages to aromatics is also shown. Tabulated values of each area can be found in Table S2.               

produced the lowest average $CO_2$ flux while lodgepole needle decomposition released the most $CO_2$. The shaded control added in 2018 produced $CO_2$ magnitudes that aligned or were less than the exposed control, confirming shading effects from the needles did not drive the differences observed in the needle samples (Figs. 3A and 3B).

**Table 1 Deionized water extractable nutrients as a function of needle type for harvested needles.**

|  | DOC (mg-C/g dry litter) | TN (mg-N/g dry litter) | SUVA (g dry litter/mg-m) |
|---|---|---|---|
| Impacted Spruce | 3.79 (±0.34) | 0.049 (±0.003) | 84 (±2) |
| Healthy Spruce | 2.29 (±0.15) | 0.041(±0.001) | 130 (±6) |
| Lodgepole | 2.16 (±0.48) | 0.021 (±0.002) | 146 (±28) |

Notes:
Results are normalized based on an air-dried mass basis.
Each parameter averaged across triplicates ($n$ = 3).
Standard deviation shown in parenthesis.

Seasonal trends were also observed during sampling events. Gas flux measurements were influenced by soil moisture as evidenced by peaks in $CO_2$ production that followed peaks in soil moisture (Fig. 3). This observed relationship between $CO_2$ production and soil moisture resulted in a positive Pearson correlation in 2018 for all sample types (average $P$ = 0.01, $R^2$ = 0.95); however, no significant correlation was determined for the year 2017. In August 2017, peaks in $CO_2$ production for the lodgepole (5,600 ± 1,100 mg-C $m^{-2}$ $d^{-1}$) and healthy spruce needles (5,400 ± 500 mg-C $m^{-2}$ $d^{-1}$) were almost two-fold higher than the needle-free control (3,200 ± 500 mg-C $m^{-2}$ $d^{-1}$). This peak in production was followed by a return to baseline conditions by the next sampling event (Fig. 3A). An analogous $CO_2$ peak was observed in 2018 after a moisture event in late June despite overall drier annual soil conditions. Interestingly, during the two high moisture time points, the decomposition of healthy spruce needles released more $CO_2$ than impacted spruce in 2017 (5,400 ± 500 mg-C $m^{-2}$ $d^{-1}$ vs 4,100 ± 500 mg-C $m^{-2}$ $d^{-1}$ respectively); however, this trend was muted and potentially reversed in 2018 (2,200 ± 400 mg-C $m^{-2}$ $d^{-1}$ vs 2,800 ± 200 mg-C $m^{-2}$ $d^{-1}$, respectively). Sampling throughout 2017 and 2018 provided interesting insights related to seasonal parameters due to contrasting seasonal temperature and moisture conditions between years. The hourly soil conditions collected during each gas flux measurement averaged 14 °C, 16% moisture in 2017 and 21 °C, 10% moisture in 2018 (Figs. 3C and 3D). These observed differences in soil temperature and moisture coincide with lower $CO_2$ peaks for all sample types during the drier 2018 monsoon season.

Methane flux averaged a removal rate of −1 mg C $m^{-2}$ $d^{-1}$ above needle and control samples with no clear differences in association with needle presence in contrast to the control. Seasonal trends in $CH_4$ behaved similarly to the observed $CO_2$ fluxes in which the data points appear to correspond with soil moisture (Fig. S2). Nitrous flux was near ambient conditions with no clear trends apparent as a function of uptake or release. Ammonia was also monitored; however, no clear trends were apparent as a function of uptake or release among field triplicates.

## Soil and porewater signatures

The 2017 and 2018 porewater collections revealed over time that the healthy spruce and lodgepole needles exported higher DOC concentrations within the aqueous near-surface in contrast to the control and impacted spruce samples. No clear trends were determined

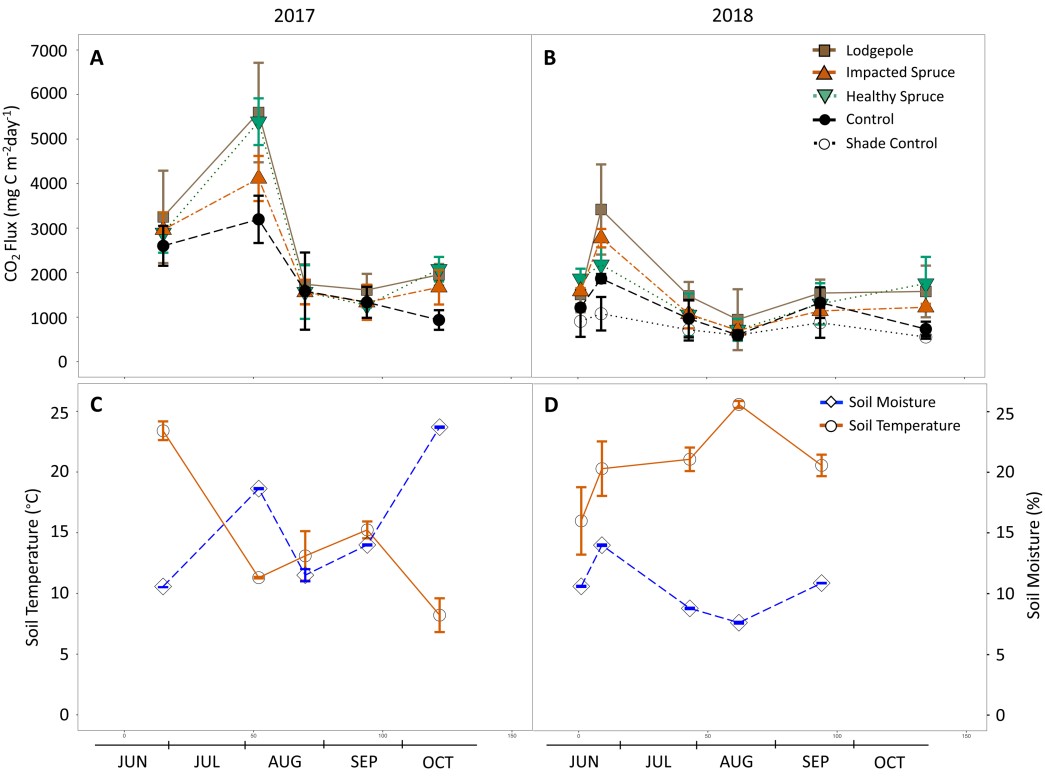

**Figure 3 Gas flux as a function of season and needle decomposition.** (A and B) Measured carbon dioxide above needle collars and (C and D) on-site soil moisture and temperature during the sampling events of 2017 (A and C) and 2018 (B and D). The gas flux error bars indicate plus or minus one standard deviation ($n$ = 3 or 4). The soil temperature and moisture error bars indicate plus or minus one standard deviation of the hourly values collected during each sampling period. Soil temperature and moisture values are not available for the final date in October 2018.

in total nitrogen concentrations during June 2017 due to high variability in lodgepole (0.38 ± 0.14 mg/L), healthy spruce (0.67 ± 0.29 mg/L) and impacted spruce (0.49 ± 0.19 mg/L). May 2018 sampling returned values below detection for all samples except for lodgepole (0.32 ± 0.30 mg/L). Late season measurements in October 2017 revealed similar concentrations between healthy spruce (0.79 ± 0.29 mg/L) and lodgepole (0.63 ± 0.24 mg/L) and lower concentrations for impacted spruce needles (0.38 ± 0.12 mg/L). Late season 2018 sampling returned similar concentrations between healthy spruce (0.92 ± 0.29 mg/L) and lodgepole (0.76 ± 0.10 mg/L) and lower concentrations released by impacted spruce needles (0.57 mg/L) however no standard deviation could be calculated due to low porewater yields. There were no clear differences in aromatic signatures (SUVA) as a function of needle presence (Table 2).

Soil extractions in October 2017 agree with the porewater results of higher soil DOC concentrations in association with needle decomposition. Interestingly, in contrast to the harvested needle water extraction analysis, soil porewater under lodgepole needles contained the highest extractable DOC compared to the spruce needles, reaching more than a two-fold higher concentration (Table 3). Overall, the total extractable nitrogen in soil under all needle samples were lower than the control; however, differences in nitrogen

**Table 2 Porewater results from four sampling events in 2017–2018.**

|  | DOC (mg/L) | TN (mg/L) | SUVA (L/mg-m) | DOC (mg/L) | TN (mg/L) | SUVA (L/mg-m) |
|---|---|---|---|---|---|---|
|  | June 2017 | | | May 2018 | | |
| Impacted Spruce | 5.3 (±1.4) | 0.49 (±0.19) | 2.2 (±0.8) | 6.6 (±2.1) | BDL | 3.0 (±0.8) |
| Healthy Spruce | 9.3 (±5.1) | 0.67 (±0.29) | 2.4 (±1.0) | 9.4 (±3.1) | BDL | 4.8 (±4.4) |
| Lodgepole | 4.3 (±1.1) | 0.38 (±0.14) | 2.1 (±0.9) | 10.3 (±3.3) | 0.32 (±0.30) | 3.7 (±0.8) |
| Control | 4.9 (±2.1) | 0.50 (±0.21) | 2.5 (±1.2) | 6.3 (±1.8) | BDL | 3.8 (±0.6) |
|  | October 2017 | | | October 2018 | | |
| Impacted Spruce | 6.7 (±2.3) | 0.38 (±0.12) | 1.85 (±0.87) | 9.4– | 0.57– | 2.1– |
| Healthy Spruce | 14.9 (±6.2) | 0.79 (±0.29) | 2.10 (±0.99) | 14.1 (±3.0) | 0.92 (±0.29) | 2.4 (±0.6) |
| Lodgepole | 12.4 (±3.4) | 0.63 (±0.24) | 1.78 (±0.54) | 16.8 (±5.9) | 0.76 (±0.10) | 4.3 - |
| Control | 11.8 (±2.0) | 0.68 (±0.04) | 1.84 (±0.60) | 13.0– | 0.958– | – |

Notes:
Each parameter was averaged across two consecutive days of sampling to increase sample size.
Standard deviation shown in parenthesis.
BDL = below detection limits.

**Table 3 Soil extractions from sacrificed single collars in October 2017.**

|  | DOC (mg/kg) | TN (mg/kg) | NO$_2$-N (mg/kg) | NO$_3$-N (mg/kg) | NH$_4$-N (mg/kg) | TON-N (mg/kg) | SUVA (kg/mg-m) | pH |
|---|---|---|---|---|---|---|---|---|
| Impacted Spruce | 63.0 | 7.63 | 0.10 | 2.41 | 0.10 | 5.02 | 0.52 | 6.04 |
| Healthy Spruce | 57.5 | 4.08 | 0.06 | 0.07 | BDL | 3.96 | – | 6.37 |
| Lodgepole | 167.2 | 5.86 | 0.08 | 0.18 | BDL | 5.60 | 0.66 | 6.02 |
| Control | 41.7 | 8.81 | 0.10 | 4.19 | 0.12 | 4.40 | 0.43 | 6.61 |

Notes:
Results are normalized based on an air-dried mass basis.
Soil samples were collected from a single collar for each sample type ($n = 1$).

speciation as a function of needle presence are apparent. The inorganic nitrogen species are low in association with needle decomposition while organic nitrogen comprised the highest percentage of the total nitrogen in soil associated with needle decomposition in contrast to the control. Specifically, organic nitrogen comprised an average of 86% of the total nitrogen in all three soil types associated with needle decomposition, while a lower percentage (50%) was observed in the needle-free control (Table 3). In addition, the needles resulted in lower soil pH values (average of 6.1 ± 0.2 across needle samples) than that of the control (pH ~ 6.6).

## Soil microbial communities in association with needle decomposition

Beta diversity differed as a function of sample type for both fungal and bacterial soil communities. Similar to gas flux, the most significant clustering determined by principal coordinate analysis was observed during or after high moisture events sampled in August 2017 and July 2018 (weighted UniFrac Adonis test: bacterial: $p < 0.05$ and fungal: $p < 0.01$; Fig. 4). Consistent with the geochemical analyses, soil communities associated with needle samples clustered separately from controls. To a lesser extent, communities under lodgepole decomposition were distinct from those under spruce needles. In contrast,

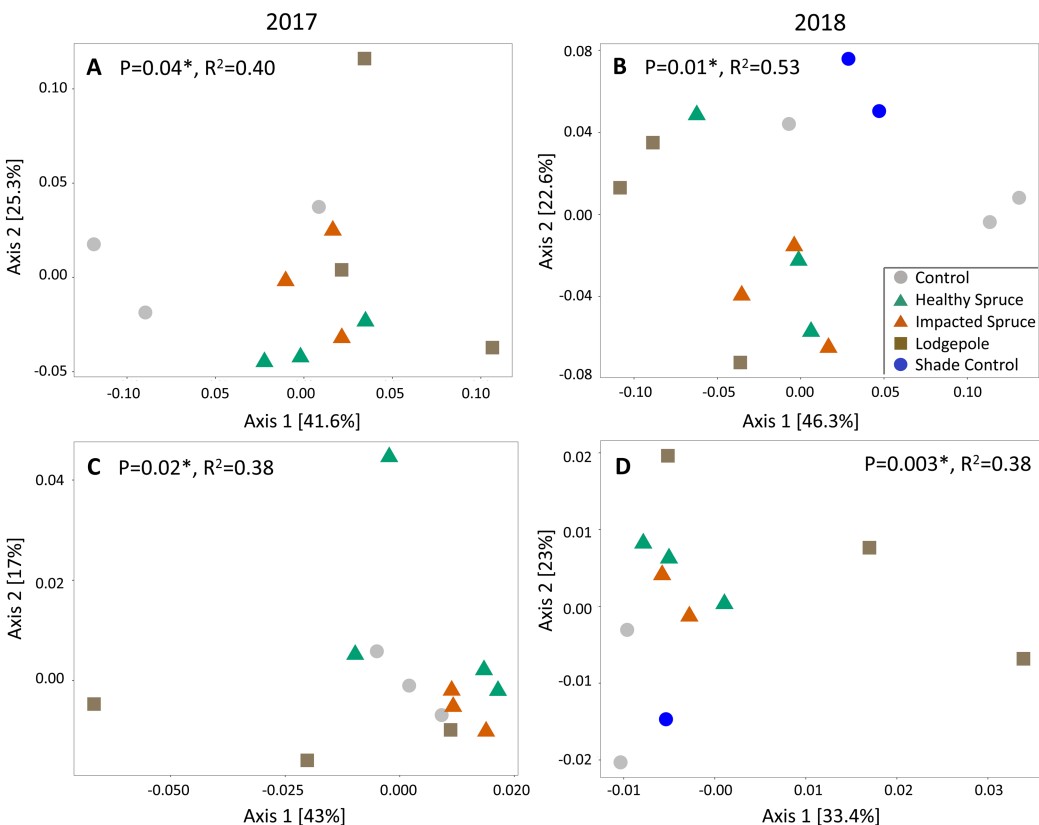

**Figure 4 Needle decomposition shifts bacterial and fungal communities during wet seasons.**
(A and B) Bacterial and (C and D) fungal communities significantly cluster based on needle type during the high moisture event of August 2017 (A and C), and after a high moisture event in July 2018 (B and D) using the weighted UniFrac distance metric. Fungal communities under the spruce samples were grouped together in the July 2018 analysis due to low sample count. $P$ and $R^2$ values for each figure are shown.

communities under the impacted spruce needles were indistinguishable from healthy spruce. Despite significant community clustering, alpha diversity metrics were not significantly impacted by the decomposition of different needle types in comparison to the control (Fig. S3).

To further investigate differences in microbial communities in soil under the needle samples, differential abundance tests were conducted on the top 15 most abundant genera during the 2017 and 2018 high moisture sampling events to determine differentiation in relative abundances at the genus level (Fig. 5). There were no significant differences between the shaded and exposed needle-free control samples or for healthy and impacted spruce. Communities under the lodgepole needles were most different from those found under the control with increasing significance in 2018 (Fig. 5; Fig. S4). Specifically, lodgepole significantly differed from the control in higher abundances of potential nitrogen fixers of the genera *Bradyrhizobium* and *Variibacter*. In addition, by comparing lodgepole and the controls, lower abundances of chemoorganotrophs *Segetibacter*, *Flavisolibacter*, *Ferruginibacter* and a potential nitrogen fixer *Massilia* were observed in the

**A**

| Phylum; Genus | AUG 2017 | | |
|---|---|---|---|
| | Control | Lodgepole | Spruce |
| Proteobacteria; Bradyrhizobium | 5.7 | 9.7 | 8.5 |
| Bacteroidetes; Segetibacter | 11.8 | 2.6 | 6.5 |
| Proteobacteria; Sphingomonas | 5.8 | 4.9 | 7 |
| Acidobacteria; Bryobacter | 3.8 | 4.5 | 6 |
| Bacteroidetes; Flavisolibacter | 5.2 | 1.7 | 3.3 |
| Verrucomicrobia; Chthoniobacter | 2.1 | 1.4 | 3.6 |
| Firmicutes; Bacillus | 2.7 | 2.4 | 2.7 |
| Proteobacteria; Variibacter | 1.3 | 3.2 | 2.7 |
| Proteobacteria; Massilia | 2.2 | 2 | 2.3 |
| Bacteroidetes; Ferruginibacter | 2.7 | 1.3 | 2.3 |
| Bacteroidetes; Flavobacterium | 0.8 | 5 | 1.1 |
| Acidobacteria; RB41 | 1.8 | 1.6 | 1.9 |
| Actinobacteria; Pseudonocardia | 1.9 | 2 | 1.7 |
| Actinobacteria; Gaiella | 1.3 | 2.3 | 1.3 |
| Actinobacteria; Blastococcus | 1.6 | 1.5 | 1.6 |

**B**

| Phylum; Genus | JUL 2018 | | |
|---|---|---|---|
| | Control | Lodgepole | Spruce |
| Proteobacteria; Bradyrhizobium *^ | 3.6 | 8.1 | 6.8 |
| Acidobacteria; Bryobacter | 6.1 | 7 | 4.5 |
| Bacteroidetes; Segetibacter **^ | 10.2 | 0.8 | 2.7 |
| Proteobacteria; Sphingomonas | 4.6 | 3.6 | 4.1 |
| Firmicutes; Bacillus | 3.7 | 3.3 | 2.7 |
| Proteobacteria; Massilia ^ | 2.6 | 1 | 3.1 |
| Bacteroidetes; Flavisolibacter **^ | 4.4 | 0.8 | 1.4 |
| Proteobacteria; Variibacter ^ | 1.3 | 3.5 | 2 |
| Actinobacteria; Pseudarthrobacter | 3.2 | 1 | 1.7 |
| Verrucomicrobia; Chthoniobacter ** | 1.2 | 1.6 | 3 |
| Actinobacteria; Blastococcus | 2.4 | 1.5 | 2 |
| Acidobacteria; RB41 | 2.2 | 1.2 | 2.1 |
| Actinobacteria; Nocardioides | 2.2 | 1.2 | 2 |
| Actinobacteria; Pseudonocardia | 2.1 | 1.2 | 1.8 |
| Bacteroidetes; Ferruginibacter ^ | 1.8 | 0.6 | 1.6 |

**Figure 5 Top 15 most abundant bacterial genera differ as a function of sample type.** Top 15 genera in (A) August 2017 and (B) July 2018. Numeric values within each box represent the percent read abundance. Spruce values are binned healthy and impacted spruce ($n = 6$). Control in July 2018 is binned shaded and non-shaded ($n = 5$). Mean abundance values are shown within each heatmap box with color representing numerical ranges visually with low values in blue and high values in red. Symbols indicate significant differences determined by differential abundance analysis as follows: ^Significant difference between lodgepole and controls, *Significant difference between impacted spruce and controls, **Significant difference between the binned impacted and healthy spruce from the controls.

lodgepole samples. Further, comparing lodgepole to the binned spruce samples, lodgepole is significantly lower in abundance of *Ferruginibacter*, *Segetibacter* and *Massilia*. Binned impacted and healthy spruce samples significantly differed from the controls with higher abundance of *Chthoniobacter* and lower abundances of *Segetibacter* and *Flavisolibacter* (Fig. 5).

Turning to Eukarya, larger variations in relative abundance were observed and taxonomic rank was not completed at the Genus or Family level. Few significant differences were determined with differential abundance tests; however, basic comparisons between samples were possible. Fungal communities followed a similar trend to bacteria in which the communities under impacted and healthy spruce needles are similar, while there are differences between lodgepole and spruce (Fig. S5). Further, based on mean relative abundances the Phylum Ascomycota comprised 50% mean abundance under the control, 52% under healthy spruce, 40% under impacted spruce, and 25% under lodgepole samples (Fig. S6). In contrast Basidiomycota represented 36% of the mean abundance under lodgepole and 10–19% under the spruce and control samples (Fig. S6).

## DISCUSSION

We gained a better understanding of the effects of needle decomposition on terrestrial soil biogeochemical processes by studying the compositional differences of senesced coniferous needles between two species and as a function of bark beetle impact. Additionally, by isolating the needle litter from the complexity of the tree canopy and rhizosphere, we were able to investigate the selective pressures derived from litter presence and decomposition over time under the same environmental conditions. By extension, this enhanced our understanding of biogeochemical cycling in natural and disrupted ecosystems. However, it is noted that conclusions with respect to bark beetle impacts are limited in our dataset to spruce trees and the study was further limited to two conifer species. Three key findings emerged from this study: (1) Under the same conditions and location, tree species exerts a larger role in needle decomposition dynamics than bark beetle impact, (2) conifer litter decomposition enhances soil organic carbon and nitrogen cycling, heterotrophic respiration, and exerts pressures on resident microbial ecology, and (3) biogeochemical signatures are enhanced during high moisture conditions.

### Needle litter chemistry

The behavior of plant litter as it decomposes can be attributed to the quality of organic matter available for microbial decomposition. The amount of bioavailable nutrients and organic matter within the needles depends on the plant phenology, which determines the physiology of carbon and nitrogen storage throughout the tree (*Millard & Grelet, 2010*). Differences in initial litter quality can affect soil chemistry and microbial communities with the potential for more microbial activity associated with higher quality litter, which can in turn affect soil C and N retention within the first year of decomposition (*Šantrůčková, Krištůfková & Vaněk, 2006*; *Fraterrigo, Ream & Knoepp, 2018*). As a result, discrete differences in organic content and nutrients as a function of needle type can shed light on the expected behavior of plant litter decomposition in the environment. To this end, high quality litter is understood to have higher nitrogen concentrations (low C:N ratios) and low recalcitrance, which can be estimated by lignin content. Following this understanding, the results of total C, N and FTIR analysis revealed that the harvested naturally senesced lodgepole needles are a higher quality substrate than naturally senesced spruce, while the beetle impacted spruce needles are of the lowest quality. A similar study comparing senesced conifer litter decomposition rates across forests within the Rocky Mountains agree with the conclusion that lodgepole contains the lowest lignin levels, but contrasts with our observation of lodgepole containing the highest C:N ratio (*Stump & Binkley, 1993*). After 2 years of decomposition, the same results by *Stump & Binkley (1993)* agree with the trend described in this study of a decrease in C:N for the spruce needles and an increase in lodgepole. The total carbon losses in this study after 2 years of decomposition revealed a modest decrease in lodgepole compared to the impacted spruce needles and a large standard deviation in the healthy spruce (Table S1). While a lack of mass balance quantification limits interpretation, possible explanations for modest carbon loss include decomposer biomass formation as well as

different mass loss rates associated with hydrolysis and oxidation due to the physical differences in needle shape and size between tree species (*Ono et al., 2011*).

FTIR analysis revealed lower quantities of lignin and higher quantities of cellulose in the lodgepole needles compared to both spruce needle types throughout 2 years of decomposition (Fig. 2). These observations indicate functional differences in the decomposition processes of lodgepole and spruce needles. This observation agrees with a litter decomposition study by *He et al. (2019)* that reported periodic lignin losses associated with increases in cellulose. Lignin and cellulose decomposition processes can be controlled by climate, litter quality, and the decomposers present. With the variability of climate and native soil microorganisms removed between samples in this study, the needle chemistry is likely the main driver of the differences observed in decomposition across needle type. Lignin and cellulose decomposition appear to be linked, which can be attributed to the organization of polymers within plant cell walls. A greater distribution of lignin can prevent decomposers from accessing more labile organics within the cell (*Berg & McClaugherty, 2014*). As a result, litter types with higher levels of lignin may exhibit reduced decomposition rates until lignin degradation occurs. This explains why the lodgepole needles behave differently from the spruce needles which contain higher lignin levels.

## Carbon & nitrogen cycling

During the process of decomposition, soil carbon and nitrogen speciation can shift as a result of abiotic and biotic factors. Specifically, chemical attributes of soils can influence decomposition processes differently across tree species (*Vesterdal, 1999*; *Berg, 2000*). By importing conifer needles to the same location, this study limited these effects to focus on the variable of needle litter chemistry. With this in mind, studies have reported similar observations across a variety of study locations which include increased inorganic soil nitrogen in association with bark beetle impact (*Clow et al., 2011*; *Brouillard et al., 2017*). Interestingly, while the harvested impacted spruce needle extractions returned the highest total nitrogen concentrations, inconsistent and contrasting trends in porewater total nitrogen and organic carbon were observed. Differences between the initial needle extractions and porewater composition over 2 years can be influenced by changes in nutrient distribution and quantity within the needles when decomposition begins (*Berg & McClaugherty, 2014*).

A study by *Kopáček et al. (2018)* observed terrestrial nitrogen speciation significantly responded to Norway spruce bark beetle tree death with increased $NH_4^+$ in the soil adjacent to tree dieback due to mineralization of organic N in the decomposing plant matter followed by nitrification. In contrast, this study revealed lower concentrations of inorganic nitrogen in soil under all needle samples when compared to the needle-free control. As a result, the fraction of organic nitrogen comprising the total extractable nitrogen in soil after 1 year of decomposition was the highest under all needle samples (Table 3). This observation of high organic nitrogen and relatively low inorganic nitrogen could be due to the dependency of nitrogen distribution on the litter C:N ratio. According to a study by *Šantrůčková, Krištůfková & Vaněk (2006)*, a litter C:N threshold value of 32 is the
point in which mineralized soil nitrogen concentrations begin to release in measurable concentrations. This threshold ratio value varies between studies, as it is dependent on the site location, litter age, and species (*Eldhuset, Kjønaas & Lange, 2017*). The C:N ratios in the needles specific to this study were above 45 during 2 years of decomposition. In addition, due to the dominance of organic nitrogen in soils associated with needle decomposition in contrast to the control, the needle chemistry appears to enhance microbial N accumulation while little N mineralization is occurring. This agrees with other results in which nitrogen mineralization was low for lodgepole and moderate for spruce during decomposition (*Stump & Binkley, 1993*).

In addition to nitrogen shifts in association with needle decomposition, the study by *Kopáček et al. (2018)* revealed increased soil DOC concentrations within the first years of beetle impacted Norway spruce tree death due to organic carbon production from dead biomass. Our results exhibit similar results of increased near-surface carbon cycling in association with seasonal $CO_2$ production and higher magnitudes of extractable DOC in soil underlying needle decomposition. The largest magnitude of $CO_2$ production was observed in association with lodgepole needles, especially during high moisture events in which respiration was almost two-fold higher than the needle-free controls. Heightened decomposition of higher quality litter leads to nutrient utilization and ultimately terrestrial microbial respiration, which in turn is expected to decompose more quickly initially (*Berg, 2000*; *Fraterrigo, Ream & Knoepp, 2018*). As a result, lodgepole forests may contribute a larger flux of $CO_2$ to the atmosphere during needle drop compared to spruce forests, especially during warmer snow-free seasons with higher soil moisture. Looking closer at healthy vs impacted spruce needles, clear differences were not determined in $CO_2$ flux magnitudes between these needle types despite chemical differences, which suggests that beetle impacted spruce needle drop alone will not shift forest soil respiration rates from that of naturally senesced spruce needles. Past studies comparing impacted forest plots to non-impacted have shown variations in results, but with a consensus of respiration values that are lower or unchanged in soil under impacted trees in comparison to soils under healthy trees for both spruce (*Speckman et al., 2015*) and lodgepole (*Brouillard et al., 2017*).

## Soil microbial selection

In conjunction with geochemical shifts, microbial community analysis provided further insights into the collective impacts of needle decomposition. Past studies have documented shifts in soil bacterial (*Mikkelson, Lozupone & Sharp, 2016*; *Mikkelson et al., 2017*) and fungal communities (*Štursová et al., 2014*) after beetle infestation. However, these observed shifts are related to variables that include litter decomposition and the cessation of rhizodeposition among other variables such the changing water and energy budgets that occur during tree mortality (*Mikkelson et al., 2011*). As significance was most pronounced near high moisture events, our results suggest that litter decomposition plays a more significant role in microbial community selection during the summer wet season when top down infiltration is occurring.

In comparing the bacterial communities, the presence of higher quality litter also exerted the strongest selective pressures on microbial communities. Interestingly, despite initial chemical differences between naturally senesced and beetle impacted spruce needles, no significant differences in microbial differential abundance were observed. This suggests that inherent complex organics associated with the tree species exert a stronger impact on biogeochemical cycling, while the impact of beetle infestation on the tree has little to no impact on the decomposition processes of spruce litter. This agrees with litter quality decomposition studies that have concluded complex organics, as well as other elements are the driving force of litter decomposition over C:N ratios (*Fogel & Cromack, 1977*; *Hobbie, 2015*).

The increased relative abundance of putative nitrogen fixers (*Bradyrhizobium* and *Variibacter*) associated with rhizospheric processes suggests that the conditions necessary for nitrogen fixation are present within the needle decomposition horizon. However, due to the lack of observed mineralization in the soil chemistry, and no predominant abundance of ammonification, nitrifying, or denitrifying bacteria, the results reveal how crucial the rhizosphere is for the complete cycling of nitrogen and further why larger impacts to the nitrogen cycle are not observed in this study. This suggests that while the needle litter decomposition contributed to carbon cycling in this study, nitrogen cycling and especially inorganic nitrogen is likely linked to other mechanisms at the tree-scale, such as belowground nutrient contributions associated with rhizodeposits.

## CONCLUSION

The results of this study shed light on the complex behavior of needle decomposition as a function of abiotic and biotic factors that are associated with temporal decomposition. Over a study period of 2 years, we have gained further insight into the expected impacts during the first stages of litterfall for healthy and beetle disrupted ecosystems. In a situation where needles fall to the forest floor following tree mortality it is likely that observable impacts will consist of increased $CO_2$ production from heterotrophic activity while enhanced inorganic nitrogen cycling will occur depending on the C:N ratio of the litter and rhizospheric processes. As more noticeable impacts on soil flux and microbial communities were observed during high moisture events, the biogeochemical signatures will be most pronounced in high moisture environments. This has been observed in past studies, in which moisture is correlated with higher coniferous decomposition rates (*Pandey & Singh, 1982*; *Gunadi, Verhoef & Bedaux, 1998*).

The expected outcomes of litter decomposition on soil biogeochemistry will differ as a function of tree species. It can be expected that lodgepole forest soil microbial communities will be different from those of a spruce forest within the first 2 years of decomposition after needle fall. During bark beetle infestation, needle decomposition is likely to exert a comparatively modest selective pressure on soil biogeochemistry when contrasted with other forest relevant processes such as evapotranspiration, canopy interception, energy fluxes, as well as rhizodeposition and the cessation of such processes.

## ACKNOWLEDGEMENTS

Field access and support were provided by the Rocky Mountain Biological Laboratory in Gothic, CO with logistical support from Jennifer Reithel. We thank Brent Brouillard, Chelsea Wilmer, Kayla Hubbard, Jake Wands, and Sabrina Nesladek for field support. Alexander Honeyman, Blake Stamps, and Gary Vanzin aided in laboratory and bioinformatic techniques, as well as all members in the Geo-Environmental Microbiology Lab at the Colorado School of Mines. We thank PeerJ editor Douglas Burns, Jenna Zukswert, and an anonymous reviewer for their insightful suggestions during the peer review processes.

### Funding

Financial support was provided by the U.S. Department of Energy (DOE), Office of Science, Office of Biological and Environmental Research under exploratory university-led research: DE-SC0016451 with support through the Lawrence Berkley National Laboratory's Watershed Function Scientific Focus Area under contract DE-AC02-05CH11231 (Lawrence Berkeley National Laboratory; operated by the University of California). The funders had no role in study design, data collection and analysis, decision to publish, or preparation of the manuscript.

### Grant Disclosures

The following grant information was disclosed by the authors:
U.S. Department of Energy (DOE), Office of Science, Office of Biological and Environmental Research: DE-SC0016451.
Lawrence Berkeley National Laboratory, University of California: DE-AC02-05CH11231.

### Competing Interests

Eoin L. Brodie and Jonathan O. Sharp are Academic editors for PeerJ.

### Author Contributions

- Laura T. Leonard conceived and designed the experiments, performed the experiments, analyzed the data, prepared figures and/or tables, authored or reviewed drafts of the paper, and approved the final draft.
- Kristin Mikkelson conceived and designed the experiments, analyzed the data, prepared figures and/or tables, authored or reviewed drafts of the paper, and approved the final draft.
- Zhao Hao performed the experiments, analyzed the data, authored or reviewed drafts of the paper, and approved the final draft.
- Eoin L. Brodie conceived and designed the experiments, authored or reviewed drafts of the paper, and approved the final draft.
- Kenneth H. Williams conceived and designed the experiments, authored or reviewed drafts of the paper, and approved the final draft.

- Jonathan O. Sharp conceived and designed the experiments, prepared figures and/or tables, authored or reviewed drafts of the paper, and approved the final draft.

## Field Study Permissions

The following information was supplied relating to field study approvals (i.e., approving body and any reference numbers):

The Rocky Mountain Biological Laboratory approved the field work.

## Data Availability

The DNA sequencing data is available at NCBI SRA: PRJNA605259. The raw geochemical datasets are available in the Supplemental Files. Written codes used to analyze the DNA sequencing data are available at GitHub: https://github.com/ltleonard/Leonard-et-al.-Needles.

## Supplemental Information

Supplemental information for this article can be found online at http://dx.doi.org/10.7717/peerj.9538#supplemental-information.

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
