# Peer review of "A comparison of lodgepole and spruce needle chemistry impacts on terrestrial biogeochemical processes during isolated decomposition"

_PeerJ, doi:10.7717/peerj.9538_

## Round 0.1 · original submission · Major Revisions

Thank you for submitting this paper for consideration to publish in PeerJ in which you compare decomposition among two coniferous species. Your paper has received two thorough reviews and the reviewers have offered some constructive criticism that will help to improve the paper.

Overall, both reviewers find this to be a well-constructed study and well-written paper that addresses an important research question. The reviewers have asked for additional information on the soils and have asked for clarification and additional insight into some seemingly conflicting results in carbon fluxes. I have marked the paper as requesting major revisions, but that is mainly a reflection of the large number of comments, and I see no reason why you should not be able to address all expressed reviewer concerns. If you are willing to re-submit a revised paper please do so along with a memo that includes specific responses to the reviewers' criticisms.

Reviewer 1 ·

Basic reporting

The manuscript text is clear, references adequate, for some comments how to possibly improve tables and figures see General comments for the author, please.

Experimental design

The manuscript brings original primary research with well defined questions. For some shortcomings in Methods and data interpretation see General comments for the author, please.

Validity of the findings

The manuscript is novel and data are robust. For more details see General comments for the author, please.

Additional comments

General comments:
The manuscript (MS) summarizes results of two-year field experiment with decomposition of spruce and lodgepole needles on chemical and microbial composition of soil. The needles were removed from forest plots and incubated under identical natural conditions on soil surface of subalpine meadow (Colorado, USA) to avoid influence of differing tree roots and rhizosphere effects. Authors evaluate effects of needle type (lodgepole vs. spruce and healthy vs. bark beetle attacked spruce) and chemical composition on CO2 (and CH4) fluxes, leaching of C and N, and soil microbial communities. The MS represents a complex study, useful for other scientists interested in biogeochemistry of forest ecosystems and modelling. However, some important information, especially details on measurements of soil moisture and temperature, and CH4 fluxes are missing. Some contrasting results (e.g., higher CO2 production and DOC leaching from lodgepole vs. spruce needles despite lower total C loss during decomposition) deserve explanations. More details, including some minor changes and specific comments how to improve the MS quality are given chronologically below:

Specific comments:
Line (L) 32 and throughout the MS: “a lower subalpine meadow”. Lower than what? Either use “a subalpine meadow” or better explain, please.
L 34 and throughout the MS: Several terms are used for needle incubations (weathering, decomposition, natural decay). Is there any reason for using synonyms? If not, the most frequently occurring “decomposition” should be used consistently.
L 61: Change “pine” to more general “conifer”.
L 69: “Fraterrigo et al., 2018.” or “Fraterrigo, Ream, and Knoepp, 2018”?
L 75: Year “2019” is missing (according to the reference; L 563).
L 81: Porewater composition?
L 83: “at a subalpine meadow environment”
L 86: Change “has since intensified since” to “has intensified since”.
L 97: “(Dendroctonus ponderosae)” repeats (see L 84).
L 105: “(Dendroctonus rufipennis)” repeats (see L 86).
L 126: Explain abbreviation “SNOTEL”, please.
Ls 128–129: There is: “Mean annual values over the study years of 2016-2018 recorded averaged temperatures of 3.9 ºC, maximum air temperature 9.8ºC, minimum air temperature -0.6ºC”. Are these maximum and minimum values means of all daily maxima and minima, respectively, observed through 2016–2018?
Ls 131–132: Unclear how was soil temperature measured. There is: “Soil temperatures recorded on-site averaged 6.7 ºC during the study period, reaching a maximum of 28 ºC and a minimum of -0.3 ºC.” Is it a temperature of soil surface measured by a thermometer without a radiation shield? Why are average and maximum soil temperatures so much higher than air temperature? Are they really expressed in °C (and not in °F)? The annual mean soil temperature of 6.7 ºC at an elevation of ~3,000 m seems to be high. What does the maximum temperature represent – an average of all daily maxima through 2016–2018 or a single value observed only once? Does mean soil temperature refer to annual or summer data? Are soil mean, maximum, and minimum temperatures computed from a dataset similar to air temperatures (i.e., based on the same number of readings per day in identical times)?
L 134: There is: “The collars were 17.78 cm in height”. The number could be safely rounded to ~18 cm.
Ls 200–201 and 222–223: “samples were … and frozen for transport where they were stored at 4 ºC until analysis”. Were they frozen until analysis or melted earlier (if yes, how long)?
L 226: “Parada et al., 2016” or “Parada, Needham, and Fuhrman, 2016”?
L 239: “by Honeyman et al. 2018” or “Honeyman, Day, and Spear, 2018”?
L 280 and further in the text: The decrease is given in percent (… by 6% and 15%), which is somewhat misleading because also original concentrations are given in %. It would be better (and easier for a reader) to provide all concentrations in mg per g, as it is in Table 1.
L 314: There is: “…with lodgepole needle decomposition releasing the most CO2.” However, this highest CO2 production from lodgepole litter contrasts with the lowest decrease in C concentrations compared to both types of spruce needles (Table S1). Why? Without explanation this disproportion could question the data coherence.
L 328–329: Similar disproportion as above. There is: “… green spruce needles released more CO2 than red spruce in 2017; however, this trend reverses in 2018”. Fig. 3A shows relatively small differences in the 2018 CO2 release from green vs. red needles. However, Table S1 shows a pronounced difference in the C loss during 2017–2018 (70 vs 32 mg/g in the green vs. red needles, respectively).
L 331: A potential problem with soil temperature again. There is: “Soil data revealed 10% lower moisture content and 12 ºC higher temperatures in the later 2018 months”. It is difficult to believe to such a high between-year (or between-season) difference in soil temperature. More details in Methods are necessary on how it was actually measured.
L 335: Not clear. How can be CH4 flux negative? Did you measure CH4 production or uptake? Define “removal rate” in Methods and add more details on how it was measured, please.
L 346: There is: “There were no clear differences in total nitrogen”, but TN was lower below the red than green spruce needles in all 3 available sets (Table 2).
Ls 351–354: The observed highest concentrations of soil extractable DOC below lodgepole needles are in contrast to the total C loss from the needles that was only 18 mg/g for lodgepole but 32–70 mg/g for spruce needles (Table S1). Why?
Ls 382–384: The sentence: “In addition … the controls” is not clear.
L 415: “(Millard, 2010)” or “(Millard and Grelet, 2010)”?
L 452: “Šantrůčková et al. in 2006” or “Šantrůčková, Krištůfková, and Vaněk in 2006”?
L 471: “Fraterrigo et al., 2018” or “Fraterrigo, Ream, and Knoepp, 2018”?
L 527: I do not understand the term “root deposition”.
L 536–537: I do not understand the terms “tree hydrology” and “rhizosphere deposition”.
L 589: The reference “D. Eldhuset T, Kjønaas OJ, Lange H. 2017” is in different format.
Ls 647–649: Is this a valid reference?
Fig. 3: (1) Legend: “Soil data was available” or “Soil data was not available”? (2) It is rather difficult to see differences in lines and points. Try to use larger points and more intensive colors (e.g., red for red spruce, green for green spruce, blue for pine, black for control). (3) Soil temperature is surprisingly high for elevation of >3,000 m. Is it actual temperature of soil surface during CO2 measurement? What were weather conditions and time (full sunshine at noon)? Were thermometers in radiation shields, or exposed to the direct full sunshine?
Table 1: Are results based on a dry weight (105 °C) or air-dried basis? Provide all data in mg/g and add to Methods that all results on litter composition are expressed on a DW or an AD basis. What does unit “g dry litter/mg*M” mean? Explain “*M”, please.
Table 2: Explain unit “L/mg-M”, please. Data on C concentrations could be safely rounded to one decimal place.
Table 3: Are results based on a dry weight (105 °C) or air-dried basis?

Supporting information: Legends are missing for all Figs.
END OF REVIEW

·

Basic reporting

Overall, this paper meets all of PeerJ's basic reporting standards. The English is clear, references are recent and appropriate, the paper is appropriately structured, raw data is completed and shared, and the study contains the results necessary to test the hypotheses.

I do have a few minor suggestions regarding the layout of the figures.

For Figure 1, I would suggest adding either a legend to B and C (the FTIR spectra), or at least mentioning which colors are which in the caption. It is implied that they'd be the same as the boxplots above, but it would be more thorough to add it for B and C.

For Figures 3 and S2, consider separating the Soil Moisture and Soil Temperature labels and making the right side of the figure its own axis. Even though these are on the same scale in terms of the number value, it is more conventional to have separate axes for separate variables. Consider also eliminating gridlines from the background of these figures.

For Figure 4, to clarify, are panels A and C from August 2017, and panels B and D from July 2018? It might be worth being more explicit about this.

For Figure 5, could you specify the units of abundance?

Experimental design

This paper does meet the aims and scopes of PeerJ and represents complete and original scientific research. The research questions are sound, and this paper does address them, and the methods are detailed.

Perhaps the most concerning gap for me in this experimental design is the lack of information on the morphology and mineralogy of the soils in this study. It makes sense to move the needles to another environment to isolate the biogeochemical role of these needles. In this case, the organic matter content and composition of the soil would clearly be different (this might be worth mentioning for transparency – if OM is low in both alpine meadows and forests in this region, perhaps it is not too important), but to what extent would the mineralogy of the soil be different? If the mineralogy is similar to what we’d expect in a lodgepole pine or spruce forest, and the only difference is lack of trees, then the interpretation that these results represent the isolated influence of litter is stronger. An easy way to address this would be to include a more detailed soils description in your Materials and Methods section, and to indicate that this soil type is similar, at least in bedrock and mineralogy, to where these needles were harvested. Interactions with the existing soil microbial community in the meadow and with the different assortment of organic compounds in the meadow is one factor also limiting the extrapolation to forests, but I believe that what you have done is reasonable, probably the best you could do to isolate litter from the forest while also keeping "field" conditions.

I would also recommend adding a bit more information regarding your gas flux measurements in the methods, in particular, mentioning perhaps when the sample was taken (time of day) and for how long, roughly, as this probably also influences the gas flux in soil.

I would also recommend considering clarifying how you ensured randomization in your study, if you can succinctly do so. How did you ensure that there would be no bias towards the edges of the collar?

Validity of the findings

This paper reports different chemical changes over time in lodgepole pine and spruce needles, and reports a greater change between species than within species that have been affected by beetle infestation, suggesting that changes in soil biogeochemistry as a result of beetle disturbance might be relatively small. This paper also demonstrates differences in soil biogeochemistry in general as a result of needle litter addition to soil (e.g., higher DOC in soils, higher proportion of organic N). Soil microbial communities also seemed more strongly defined by litter species, and there were no differences among green and red spruce needles, suggesting limited effects of beetle infestation on soil community composition as a result of chemical changes in needle litter. This paper also documents the effect of moisture on the clarity of biogeochemical signals, which is reasonable, given that water is often a necessary component of soil biogeochemical weathering reactions. These findings are supported by the data presented in this paper.

I recognize that one of the main challenges with this study is the relatively low sample sizes (n = 3). I commend you for pointing this out and not reporting significance values anyway, as that would be misleading. I think you still can and should, however, report measures of uncertainty in your results. You display this well in your figures by reporting standard deviation. I would recommend adding standard deviations or confidence intervals to your findings reported in the written text as well, to give readers a better sense to the amount of variability you had in your study.

One minor comment to aid readers in your interpretation, I would recommend clarifying in the Methods or Results section the connections you are making between the individual, chemical components that FTIR spectra detect and the molecules you believe that these represent (e.g., lignin, cellulose). You mention changes in terms of labile and recalcitrant compounds, but for an audience familiar with litter decomposition, it may be effective to tie into your explanation how your FTIR results link to these familiar carbon compounds early on and before describing these implications without this explicit context in the Discussion.

I have a few specific questions and recommendations for portions of your interpretations from this study:

Lines 431-438: Why might this be, that lodgepole pine and spruce needles behaved in different ways, chemically? Can you expand on the relevance of the He et al. 2019 work to your work? Or provide evidence of other factors that might cause this difference?

Lines 478-490: This is an interesting thought, but feels a bit speculative and requires quite a few assumptions that, as written, are not either clearly stated as assumptions or backed up by literature. It may also be important to acknowledge that you only collected senesced needles under healthy lodgepole pine trees, and the assumption that there’d similarly be little difference between litter from healthy and infected pine trees (as there was little difference between litter from health and infected spruce trees) would probably need to be spelled out here.

In general, that is another limitation of the study (testing of needles from only health lodgepole pine trees, as opposed to both healthy and infected) that might be worth addresing as a caveat in your discussion. It does not make your study less publishable, but helps identify the limits to which your conclusions of the effects of tree species ID vs. influence of beetle infestation are generalizable.

Throughout, exemplified in 506: I would be very clear about the fact that you are, for spruce, testing the impact of beetle infestation on the tree’s litter, specifically, and can only comment on effects of nutrient cycling as they relate to litter only.

Additional comments

Great job on this study; overall, I think it is fairly thorough and well researched.

Beyond what I have already expressed above, I have a few additional comments.

First, I had not come across a reference to "litter weathering" before reading this paper. I see what you mean, and see now that it is a term that is sometimes (rarely?) used, but I think you are capturing biogeochemical effects of litter decomposition, and so it is appropriate and potentially less misleading to be consistent about using the term "decomposition" instead of "weathering".

Second, this is very minor, but "red spruce" is also the common name of another spruce species, and so the use of "red spruce" to refer to needles that come from beetle-infested trees was a bit confusing at first. I wonder if "healthy" and "infested" might be better? Or is there is another set of terms that would be better? I understand the reasons for 'green' and 'red' given how these trees are typically referred to when they are infested... this again is a minor comment.

Third, for your Results, I notice that you often repeat your methods in a sentence or two before stating your results (e.g., lines 267 to 272 feel like methods). I do not think it is necessary to do this. Instead, I would try to emphasize your results as soon as you can, and maybe try to weave these methodological details into that sentence, or omit them altogether (since this all should be clear in your methods).

Line 473: I recommend clarifying what you mean by a "significant shift" . Significant shift in biogeochemistry?

Minor Line by Line Comments that do not require a response:

Line 129 and beyond: Add a space between the number and the unit (e.g., 73 cm instead of 73cm)

Line 130: Omit the space between / and October (September/October)

Line 227: add "by" after "as done"

---

## Round 0.2 · accepted · Accept

One of the previous reviewers examined the revision of your paper and concluded that you did a good job of addressing the previous comments and criticisms. I agree with this reviewer's conclusion.

·

Basic reporting

Overall, this paper meets all of PeerJ's basic reporting standards. Changes to references that I overlooked were caught by the other reviewer and addressed by the authors, and my earlier suggestions regarding the layout of the figures were accepted.

Experimental design

This paper does meet the aims and scope of PeerJ and represents complete and original scientific research. Modifications made by the authors to address comments that I and the other reviewer made help further clarify the experimental design and research; the manuscript has improved.

Validity of the findings

The modifications that the authors made in order to address my concerns and the other reviewer's concerns regarding interpretation of the findings were addressed. Findings are valid and reasonable, and interpretations are sound.

Additional comments

My only comment might be, in Line 506, to consider referring to the variability of climate and native soil microorganisms as "reduced" rather than "removed" in this study, as soil communities can be quite heterogeneous, even within very small areas.

Otherwise, the revision in response to the two reviews was very thorough and the manuscript improved. Great work!